# The Psychometric Properties of the DASS-21 and Its Association with Problematic Internet Use among Chinese College Freshmen

**DOI:** 10.3390/healthcare11050700

**Published:** 2023-02-27

**Authors:** Cui-Hong Cao, Chang-Yan Dang, Xia Zheng, Wang-Guang Chen, I-Hua Chen, Jeffrey H. Gamble

**Affiliations:** 1School of Foreign Languages, Shandong Women’s University, Jinan 250300, China; 2Mental-Health Education Center, Nanchang University, Nanchang 330031, China; 3School of Administration, Guangdong Polytechnic Normal University, Guangzhou 510665, China; 4Chinese Academy of Education Big Data, Qufu Normal University, Qufu 273165, China; 5Department of English, National Changhua University, Changhua 50007, Taiwan

**Keywords:** psychometrics, problematic Internet use, the DASS-21, Chinese freshmen, COVID-19 pandemic

## Abstract

During transitional periods, college freshmen may experience mental health issues. The Depression, Anxiety, and Stress Scale—21-item version (DASS-21) is commonly used for mental health assessment in China. However, evidence is lacking regarding its applicability with freshmen as a demographic. Debates also exist regarding its factor structure. This study aimed to evaluate the DASS-21′s psychometric properties with Chinese college freshmen and investigate its association with three kinds of problematic Internet use. A convenience sampling method was used to recruit two samples of freshmen—one of 364 (female 248; mean age 18.17 years) and the other of 956 (female 499; mean age 18.38 years) participants. McDonald’s ω and confirmatory factor analysis were conducted to evaluate both the scale’s internal reliability and construct validity. The results indicated acceptable reliability, with a one-factor structure inferior to a three-factor structure in terms of model fit. Furthermore, it was demonstrated that problematic Internet use was significantly and positively associated with depression, anxiety, and stress among Chinese college freshmen. Based on the prerequisite of measurement equivalence across two samples, the study also found that freshmen’s problematic Internet use and psychological distress were likely to be affected by the strict measures put in place during the COVID-19 pandemic.

## 1. Introduction

Adolescence is a key period when many psychological disorders can occur, and, broadly speaking, college life can be considered the final stage of adolescence for students [1]. Previous studies have demonstrated that, due to feeling frustrated by homesickness, academic performance, pressure to succeed, and environmental changes, as well as being in a transitional period of life, college students are likely to experience more negative emotions and mental disorders in their freshman years [2,3,4,5,6]. It was reported that college freshmen have a higher level of psychological distress, most likely stemming from experiencing the unknown, newly moving away from home, over-expectations of embarking on a new life path, or feeling overwhelmed by coursework and unfamiliar teaching methods [7]. For all these reasons, college students’ psychological distress deserves special attention [1], as it is not only associated with their struggles in academic performance, dropout rate, and excessive alcohol intake, but also negatively affects their professional development [8]. Therefore, early detection, prevention programs, and mental health interventions are necessary and can have a significant, long-term impact. As such, it is necessary and important to be able to use a reliable and valid instrument to assess the levels of freshmen’s psychological distress.

## 2. Literature Background

### 2.1. The Depression, Anxiety, and Stress Scale

The Depression, Anxiety, and Stress Scale (DASS), is a commonly utilized instrument for screening mental health problems, the full version consisting of 42 items, and the short version consisting of 21 items [9]. Developed by Lovibond et al., this scale takes the form of a self-report questionnaire, with the original purpose of having a consistent measurement system which could be used to distinguish between and define depression, anxiety, and stress, in order to assist with clinical diagnosis using additional psychometric indicators and provide a rapid and accurate tool for screening research subjects [10]. The Depression, Anxiety, and Stress Scale—21-item version (DASS-21) adopted in the present study has the advantage of being able to be completed and scored more quickly.

The DASS-21 has been empirically validated with college students from diverse cultures with high internal consistency [2,11,12,13,14,15]. In China, several studies have been conducted to assess its psychometric properties. Gong et al. first introduced the Chinese version of this scale. They conducted a survey of college students, finding that the DASS-21 had stable psychometric properties and could successfully reflect the psychological distress status of Chinese college students [10]. Another study also confirmed the satisfactory internal consistency indices for the three subscales—depression, anxiety, and stress (with Cronbach’s αs of 0.83, 0.80, and 0.82, respectively)—and a total internal consistency of 0.92 [16].

Although this evidence shows that the Chinese DASS-21 can be a useful tool for measuring the psychological distress of college students, there has not been a psychometric evaluation of this scale specifically for college freshmen. Due to the fact that this demographic in particular may be suffering from mental health difficulties, it is important to be able to screen their mental health situations with a reliable scale in order to suggest effective intervention measures [17]. Furthermore, even though there have been many consistent findings regarding the scale’s internal consistency regarding college students, many inconsistent findings have also been found regarding its factor structure (i.e., the use of a one-factor or a three-factor structure) when used with the same demographic [11,18,19]. Based on this gap, our primary aim was to evaluate the psychometric properties of this commonly used instrument—the DASS-21—for Chinese college freshmen.

### 2.2. The Association of the DASS-21 with Problematic Internet Use

In addition to validating the DASS-21 in the context of Chinese college freshmen, the association of this scale with problematic Internet use (PIU) was also investigated. Existing studies have found that PIU is strongly linked to depression, anxiety, and stress [20]. Recently, Internet and mobile technology have developed rapidly and become more widespread, as modern life has become increasingly inseparable from the Internet. Based on the 50th China Statistical Report on Internet Development released by the China Internet Network Information Center, the number of Chinese Internet users reached 1.05 billion in June 2022, with cellphone users comprising 99.6 percent of this number [21]. Despite the many advantages of using the Internet (e.g., increased accessibility for communication, information searching, and recreation) [22], the fact that people can now easily obtain access to the Internet and can afford to use it globally contributes to the rising incidence of PIU [23,24,25], a mental disorder manifested as a result of being excessively engaged in out-of-control Internet use which is detrimental to one’s well-being [26]. Adolescents experiencing negative emotions and mental disorders have a tendency to resort to the Internet in order to escape from the anxieties or difficulties they face in their daily life [23]. However, being absorbed by the Internet makes it more likely for them to experience both physical and psychological disorders [27,28], including insomnia, low self-esteem, depression, anxiety, and stress [20,29]. These physical and mental health problems are more often found among students experiencing problematic smartphone use (PSU) [30], problematic social media use (PSMU) [31,32], and problematic Internet gaming (PG) [33,34].

As technological devices have become more prevalent over the past few decades, many college students are now regularly using Internet-based applications and portals [35]. Currently, smartphones are the most commonly used device for accessing the Internet due to its multi-functionality and consequent global popularity [36]. Middle school and college students are the primary age groups of Internet users [37]. Smartphones are a convenient tool that allows people to communicate, socialize, seek entertainment, and enhance their productivity [38]. However, using smartphones excessively may have adverse effects, such as the development of PSU [36]. An analysis of 117 studies concluded that PSU is linked to psychological distress symptoms [39]. Currently, there is a debate within the literature regarding whether this is a result of being addicted to the device itself (e.g., a smartphone) or to the content and applications available and accessible on these devices (e.g., websites, applications, and social networks) [38]. One study has suggested that applications on smartphones are the source of the problematic behavior [40]. Meanwhile, using a single cross-sectional study, Stuart et al. concluded that the level of addiction was greater for the device as a whole than for each particular service it provided [38]. Therefore, the influence of PSU on the emotions of college freshmen should be studied.

In addition to smartphones, social media also exerts a great influence on people’s living habits. Social media is defined as “Internet-based channels that allow users to opportunistically interact and selectively self-present, either in real-time or asynchronously, with both broad and narrow audiences who derive value from user-generated content and the perception of interaction with others” [41]. PSMU, however, is characterized by the compulsive or poorly controlled use of social media that negatively impacts personal and professional functioning, with growing concern about PSMU among adolescents [42]. Several previous studies found that the level of PSMU is positively associated with the level of psychological distress experienced [32,43].

Besides smartphones and social media, Internet gaming is another relatively popular form of entertainment about which several public health concerns have been raised. Internet gaming disorder [44] was first treated as a mental health disorder in 2013 by the American Psychiatric Association [45]. Evidence has demonstrated that levels of anxiety, depression, and stress are significantly correlated with PG tendency [46,47].

Considerable research has been conducted aiming to detect the adverse effects of PSU, PSMU, and PG on college students’ psychological well-being. One study with Hong Kong University students as research subjects found that PSMU and PG had positive associations with psychological distress [23]. To the best of our knowledge, only one research study has been conducted to investigate the correlation among three PIUs and psychological distress, and this study concerned Malaysian university students [29]. Therefore, we note a research gap in that PSU, PSMU, and PG have rarely been examined and compared within a single study, specifically concerning college freshmen in a Chinese context. To address this gap, the second aim of this work was to assess and compare the associations between the three PIUs and psychological distress among Chinese college freshmen.

### 2.3. The Impact of COVID-19

The impact of the novel coronavirus (COVID-19) pandemic on the mental health of college freshmen cannot be ignored. Since the pandemic broke out in December 2019, the Chinese Government has enacted strict prevention and control measures. An early notice issued by the Ministry of Education of the People’s Republic of China clarified that, in the case of an infection, campuses were expected to respond quickly to follow emergency plans, which included the quick implementation of isolation protection, on-site disinfection, dormitory lockdown, and online teaching [48]. Two years later, the Omicron variant emerged, with increased transmissibility. To care for the lives and health of teachers and students, as well as to increase campus safety and the overall stability of the education system, as a general policy, China’s national education system has unwaveringly adhered to the “dynamic zero COVID” and “external defense import, internal defense rebound” policies [49]. At the time of writing, many universities continue to be under lockdown conditions and students are asked to stay on campus and even remain in their dorms in order to prevent infection and limit the spread of the pandemic. Given that various studies have found that the COVID-19 pandemic and social isolation policies may have resulted in the increased prevalence of serious mental health problems [50,51], it is important to investigate whether differences exist between freshmen’s psychological states in 2020 and those of freshmen in 2022.

To summarize, the present preliminary study had two research questions (RQs):

RQ1: What are the DASS-21’s psychometric properties regarding Chinese college freshmen, including its internal reliability and construct validity (i.e., factorial validity, convergent validity, and discriminant validity)?

In order to find the answer to this question, the present study conducted a cross-sectional study of two groups of college freshmen (i.e., freshmen in 2020, the early stage of the pandemic; and freshmen in 2022, a more recent stage of the pandemic). In addition, a multiple-group analysis was carried out to determine whether or not there was measurement invariance across these two samples.

RQ2: What is the association between the DASS-21 and PSU, PSMU, and PG?

The present study hypothesized that associations would exist between psychological distress and the three PIUs in Chinese college freshmen. Furthermore, the study expected to find different results regarding mental health between the two freshmen groups as they had experienced different stages of the pandemic.

It was decided that the combination of validating the instrument, analyzing the influencing factors (i.e., PSU, PSMU, and PG), and comparing freshmen’s psychological states between the 2020 and 2022 cohorts could provide useful insights to both policymakers and university staff to help them better understand the psychological states of those freshmen confronted with public health emergencies, empowering them to develop effective prevention and intervention measures. The study results can also be used as a reference for the formulation and adjustment of campus pandemic prevention and control policies.

## 3. Materials and Methods

### 3.1. Participants and Procedure

A total of two sets of data were collected, the first in October 2020, and the second in October 2022. The study participants were recruited utilizing convenient sampling. The selection criteria for participants included (1) being a freshman student, (2) understanding simplified Chinese, and (3) owning a smartphone with Internet access for more than three months. The demographic characteristics of the two samples are presented in Table 1. A background questionnaire was included in the survey, as well as the Chinese versions of the following three questionnaires: the DASS-21, the Smartphone Application-Based Addiction Scale (SABAS), the Bergen Social Media Addiction Scale (BSMAS), and the nine-item Internet Gaming Disorder Scale—Short Form (IGDS-SF9). The background information section included items relating to age, gender (male or female), department (science, literature, management, and other), and study program level (undergraduate or junior college). The study obtained approval from the Institutional Review Board of the Jiangxi Psychological Consultant Association (IRB ref: JXSXL-2021-J99).

The first set of data, collected in October 2020, was primarily from the city of Nanchang, in the province of Jiangxi. The authors contacted psychology teachers in colleges and universities, letting them know about the purposes of the research and the estimated time needed to answer all the questions on the questionnaire by telephone. After receiving a positive response, the researcher sent the psychology teachers a hyperlink and QR code to access the measurement materials. These teachers shared the link and QR code with their students in the classroom, either before or after class, and students completed the questionnaire immediately after receiving it. During the data collection period, universities were busy adjusting teaching after experiencing the sudden impact of the pandemic in the first half of the year, so the data collection process encountered difficulties, and only 364 participants were recruited, of which 116 (31.9%) were male and 248 (68.1%) were female, the latter being more than double the number of the former. The participants’ average age was 18.17 (*SD* = 0.42) years. All were from undergraduate colleges, studying one of the four categories of majors: literature (33%), management (14%), engineering (39.3), or “other” (13.7%).

To collect the second set of data, the author sent an invitation message to research assistants and instructors, distributed an online link to the survey, and also provided information regarding the purposes of the research and the estimated time needed to complete the questionnaire. Several research assistants and faculty members sent the hyperlink and QR code onwards, providing access to the questionnaire to potential participants. The second set of data was collected in October 2022, from schools in a total of 13 provinces, including Shanxi, Guangdong, Guizhou, and Shandong, among others. A final total of 956 participants were recruited in the second round of data collection, of which 457 (47.8%) were male and 499 (52.2%) were female, representing a more balanced distribution. The participants’ average age was 18.38 (*SD* = 1.11) years, and two thirds of the participants came from undergraduate colleges, with one third coming from junior colleges. The students reported that they majored in science (15.8%), literature (15.9%), management (16.3%), engineering (28.7%), or “other” (23.3%).

### 3.2. Measures

#### 3.2.1. The Depression, Anxiety, and Stress Scale—21-Item Version (DASS-21)

Developed by Lovibond and Lovibond, the DASS-21 is a shortened version of the original 42-item scale [9]. The scale consists of three self-reported subscales for depression, anxiety, and stress that are intended to measure the psychological distress experienced by the participant over the previous week. Each subscale comprises seven items. An example of a depression subscale item is “I found it difficult to work up the initiative to do things”. An example of an anxiety subscale item is “I was worried about situations in which I might panic and make a fool of myself”. An example of a stress subscale item is “I felt that I was using a lot of nervous energy”. The 21 items are rated on a Likert scale with four points, ranging from 0 (“did not apply to me at all”) to 3 (“applied to me very much or most of the time”). An overall higher score is indicative of a higher severity of psychological distress. Previous research has already proven that the DASS-21 possesses satisfactory psychometric properties coupled with a high degree of consistency and a confirmed unidimensional structure [7,29,52,53,54,55]. Moreover, it has been demonstrated that the Chinese version of this instrument has good internal consistency (α = 0.89) among Chinese college students [10]. Please refer to the Results section, below, for information on the scale’s reliability and validity for the college freshmen participating in this study.

#### 3.2.2. The Smartphone Application-Based Addiction Scale (SABAS)

The SABAS was adopted to gauge the severity of problematic smartphone use by college freshmen. Developed by Csibi, Demetrovics, and Szabo [56], the scale contains six items. The six items are rated on a six-point Likert scale ranging from 1 (“strongly disagree”) to 6 (“strongly agree”). An example of an SABAS item is “If I cannot use or access my smartphone when I feel like, I feel sad, moody, or irritable”. The maximum score is 36, with a higher score indicating that the participant is more severely addicted to smartphone use. Acceptable psychometric properties for this scale have been demonstrated in different languages in several previous studies, including English [56], Hungarian [57], Persian [58], and Italian [59]. The unidimensional structure SABAS’s Chinese translation has good internal reliability when used for Hong Kong University students (α = 0.75) [35] and mainland Chinese primary school students (α = 0.81) [22]. In the current study, the internal consistency of the SABAS had a McDonald’s ω of 0.745 for the first set of data and 0.884 for the second set of data.

#### 3.2.3. The Bergen Social Media Addiction Scale (BSMAS)

The BSMAS was used to assess the severity of college freshmen’s PSMU. Developed by Andreassen et al. [31], this scale contains six items, and each evaluates a person’s experience of using social media during the past year. An example item is “How often during the last year have you become restless or troubled if you have been prohibited from using social media?” The six items are rated on a five-point Likert scale with a maximum score of 30, with categories ranging from 1 (“very rarely”) to 5 (“very often”). A higher total score means that the respondent’s risk of PSMU is more severe [60]. The Chinese version of the BSMAS has been previously validated for use on Chinese primary school students with acceptable psychometric properties [22]. In the present study, the internal consistency of the BSMAS had a McDonald’s ω of 0.782 for the first set of data and 0.866 for the second set of data.

#### 3.2.4. The Internet Gaming Disorder Scale—Short Form (IGDS-SF9)

The current study adopted the IGDS-SF9 to assess the severity of PG in college freshmen. This measure is a self-report scale developed by Pontes and Griffiths based on DSM-5 criteria [61]. It comprises nine items evaluating how an individual has recently experienced gaming activity. An example item is “Do you feel the need to spend an increasing amount of time engaged in gaming in order to achieve satisfaction or pleasure?” A five-point Likert scale is used to rate the nine items, ranging from 1 (“never”) to 5 (“very often”). The sum of the individual item scores, with a minimum potential score of 9 and a maximum possible score of 45, is used to determine the overall result, with higher total scores indicating that the respondent has a more severe degree of gaming disorder. Previous research has proven that the IGDS-SF9 is a valid and reliable scale which can be used to measure PG [62]. Previous studies have been conducted to assess the Chinese version among Hong Kong adults [63], Hong Kong university students [35], and mainland Chinese primary school students, all with acceptable psychometric properties [22]. In the current study, according to the preliminary analysis, the internal consistency of the IGDS-SF9 had a McDonald’s ω of 0.906 for the first set of data and 0.930 for the second set of data.

### 3.3. Data Analysis Strategy

The current study adopted descriptive statistics and Pearson correlations for the purpose of analyzing the mean (*SD*) of the included variables and the correlations among them. Additionally, to compare how much the variables varied between the participants of the 2020 and 2022 cohorts, an independent *t*-test method was adopted in order to determine whether the discrepancies between the variables were at a statistically significant level. Subsequently, McDonald’s ω and confirmatory factor analysis (CFA) were both utilized to assess the DASS-21’s internal consistency and factorial validity. Regarding its factorial validity, two fits of structures were compared: one was a one-factor structure and the other was a three-factor structure, in addition to evaluating the individual factor structure used. It is noteworthy that we conducted an Exploratory Factor Analysis (EFA) on two separate samples. This was done to address the ongoing controversy regarding the dimensional structure of DASS-21; the EFA, which allows for the cross-loading of items, provided additional support for the use of Confirmatory Factor Analysis (CFA). However, considering the extensive testing that has been performed on the DASS-21, as mentioned previously, and that EFA is usually employed in the scale development stage [64], the EFA results are presented in the Appendix A file which accompanies the present work.

We further tested the convergence, discriminant validity, and measurement invariance using CFA. Based on the standardized factor loadings, the composite reliability (CR) and average variance extracted (AVE) were computed to investigate convergent and discriminant validity. In order to evaluate DASS-21’s measurement invariance across the two samples, a comparison of several models was conducted, which included (a) a configural model (i.e., no constrained model, as a baseline model); (b) a factor-loading constrained equal model (less constrained model); and (c) a factor-loading and item-intercept constrained equal model (more constrained model). When the differences of the fit indices from different models do not violate the criterion, the equivalence of the DASS-21 across distinct samples can be determined. Finally, for the examination of the associations between PIU and the DASS-21, structural equation modeling (SEM) was utilized in the current study to test the influence of the three PIUs on freshmen’s DASS-21 score, with gender as the controlled variable, considering potential differences in the prevalence of PIU [19,33] between males and females.

In terms of the criteria of the above-mentioned statistics, several indices were used. For factorial validity and SEM, indices including the comparative fit index (CFI), non-normed fit index (NNFI), root mean square error of approximation (RMSEA), and standardized root mean square residual (SRMR) were selected. In order to meet the criteria, it was essential to meet the following values: the values of CFI and NNFI should not be lower than 0.95; the values of RMSEA should not be higher than 0.06; and the values of SRMR should not be higher than 0.08 [65]. As for the model selection of the factor structure, the Akaike information criterion (AIC) was adopted by the current study to compare the models with acceptable fit indices to determine which had the best model fit (i.e., smaller AIC indicates a better fit) [66]. For better convergent and discriminant validity, convergent validity is considered to be satisfactory when the CR is greater than 0.70 and the AVE is greater than 0.50 for each construct [65]. Additionally, the square root of the AVE needs to be greater than the correlations between the constructs, thus supporting the discriminant validity of the construct [61]. Regarding measurement invariance, a comparison of the models using CFI, RMSEA, and SRMR was performed to verify whether measurement invariance could be supported, with the following requirements: an ΔCFI higher than −0.01, an ΔRMSEA lower than 0.015, and an ΔSRMR lower than 0.03 (for factor-loading constrained) or lower than 0.01 (for item-intercept constrained) [67].

## 4. Results

### 4.1. Descriptive Statistics and Pearson Correlations

The mean values of Sample 1 (collected in October 2020) and Sample 2 (collected in October 2022) for the observed variables and the correlations between these variables are presented in Table 2. The results show that college freshmen in 2022 had a significantly higher prevalence of depression than college freshmen in 2020 (*t* = 2.89, *p* < 0.01, Cohen’s *d* = 0.18, with the value of 0.18 indicating a small effect size). Moreover, freshmen in 2022 also showed significantly more severe PIU than freshmen in 2020, with the differences having either small effect size, small to medium effect size, or medium to large effect size, depending on the type of PIU (PSU: *t* = 2.52, *p* = 0.01, Cohen’s *d* = 0.16; PSMU: *t* = 4.66, *p* < 0.01, Cohen’s *d* = 0.29; PG: *t* = 7.88, *p* < 0.01, Cohen’s *d* = 0.49). In terms of the Pearson correlations, significant positive relationships were found among all variables. Of these coefficients, strong associations were found regarding the three emotional disorders (Sample 1: *r* = 0.59 to 0.74, all *p* < 0.01; Sample 2: *r* = 0.79 to 0.83, all *p* < 0.01). Despite the fact that the three types of PIU were mutually correlated in the two samples, we noticed that a relatively moderate magnitude of correlation was found for the three PIUs in Sample 1 (*r* = 0.29 to 0.47, all *p* < 0.01), with a relatively higher magnitude of correlation for Sample 2 (*r* = 0.50 to 0.61, all *p* < 0.01). Additionally, the three emotional disorders and three different types of PIU were all positively correlated, and paired correlations were higher in Sample 2 (*r* = 0.42 to 0.48, all *p* < 0.01) than in Sample 1 (*r* = 0.22 to 0.38, all *p* < 0.01), which is similar to the pattern seen in the association among the three PIUs.

### 4.2. Reliability, Factorial, Convergent, Discriminant Validity, and Measurement Invariance

For Sample 1 data, the internal consistency was acceptable, and the McDonald’s ωs for depression, anxiety, and stress were 0.789, 0.633, and 0.755, respectively. For the Sample 2 data, the internal consistency was excellent, and the McDonald’s ωs for depression, anxiety, and stress were 0.887, 0.825, and 0.861, respectively.

In terms of the factorial validity of the factor structure, the CFA results in the current study demonstrated that the scale fit the factor structure well in both samples (i.e., the fit indices mostly met the criteria (see Table 3)). Furthermore, the fit was significantly better for the three-factor structure than for the one-factor structure, due to a much lower AIC for the three-factor structure (Sample 1: one-factor structure = 652.67 and three-factor structure = 571.79; Sample 2: one-factor structure = 1043.83 and three-factor structure = 826.94).

As for convergent validity, the results demonstrated that the convergent validity of the DASS-21 was ideal in Sample 2, since the CR for each factor was higher than 0.70 (the CRs of depression, anxiety, and stress in the current study were 0.94, 0.78, and 0.90, respectively) and the AVE of each factor was higher than 0.50 (the AVEs of depression, anxiety, and stress in the current study were 0.70, 0.57, and 0.59, respectively). Compared to Sample 2, although the CRs of all three dimensions were also higher than 0.70 in Sample 1 (the CR of depression, anxiety, and stress in the current study was 0.89, 0.78, and 0.82, respectively), but the value of the AVEs (the AVE of depression, anxiety, and stress in the current study were 0.54, 0.34, and 0.41, respectively) was unsatisfactory. As for the discriminant validity, the results revealed poor discriminant validity for both samples for the three DASS-21 factors because the square root of the AVE for each factor was less than the correlation between the latent factors (see Table 4). These findings are further supported by the results of the EFA. The Scree plot from the EFA results shows that the number of factors extracted from both Sample 1 and Sample 2 does not align with the initial expectations of the three factors (as depicted in Appendix A). Additionally, the factor loadings reveal cross-loadings for some of the items (see Appendix A), indicating a potential issue with the overly high correlations among the latent variables in the DASS-21. These findings suggest that these three factors should be considered as an overall construct, namely, psychological distress, rather than three separate constructs.

Regarding the examination of measurement invariance across the two sets of data, the results indicated that the equivalence of the DASS-21 was supported across the two samples in terms of both factor loadings and the item thresholds in both the one-factor structure and three-factor structures (see Table 5). Specifically, the DASS-21 had generally acceptable fit indices in the configural model, with the exception of SRMR, which had a value slightly higher than 0.80. Moreover, the results of the model comparison showed that the ΔCFI ranged from −0.006 to 0.001, ΔRMSEA ranged from 0 to 0.011, and ΔSRMR ranged from 0.003 to 0.011. All these values met the criteria for measurement invariance. 

### 4.3. Associations between PIU and the DASS-21

For the purpose of examining the associations between PIU and the DASS-21 in SEM, we used the mean score of the three sub-scales (depression, anxiety, and stress) as the indicators of a general construct (i.e., psychological distress), given that the above results indicated that the score of the DASS-21 should be considered as an overall variable, due to poor discriminant validity for the three sub-factors. Using gender as the controlled variable, the results of the current study revealed that the model in both samples had generally satisfactory fit indices, despite the fact that RMSEA did not meet the criterion in Sample 1 (see Figure 1). Subsequently, the results of the path coefficients (also in Figure 1) indicated that higher PSU and PG contributed to a more severe level of psychological distress in both samples (Sample 1: PSU = 0.37, *t* = 3.83, *p* < 0.01; PG = 0.15, *t* = 2.55, *p* = 0.01) (Sample 2: PSU = 0.32, *t* = 6.74, *p* < 0.01; PG = 0.24, *t* = 5.42, *p* < 0.01). Finally, regarding the association between PSMU and psychological distress, the results showed that, although a higher PSMU was more indicative of poorer mental health in Sample 2 (β = 0.13, *t* = 2.66, *p* < 0.01), this association did not exist in Sample 1. Clearly, higher PSU and PG values reflected a more severe level of psychological distress in freshmen compared to PSMU.

## 5. Discussion

In consideration of the unique psychological status of college freshmen and the current lack of effective measurement scales for use regarding this demographic, the current investigation evaluated the reliability and validity of the DASS-21 for use among Chinese college freshmen using two sets of sample data (from October 2020 and October 2022). In addition, due to the fact that some debate still exists regarding its factorial validity, the DASS-21 factor structure was also examined in more detail. Furthermore, this study assessed the associations between three PIUs (PSU, PSMU, and PG) and freshmen’s psychological distress. The results indicated that the DASS-21 possesses robust psychometric properties for use among Chinese college freshmen and had reasonable factorial validity using either a three-factor structure or a one-factor structure, which is inconsistent with the results of previous studies [10,12,68,69]. Moreover, it was also found that PIUs had a significant and positive relationship with Chinese college freshmen’s psychological distress: the higher the PIU level, the more severe their psychological distress. PSU and PG, two out of the three PIUs, contributed more to harming the mental health of freshmen than PSMU, the third type of PIU.

### 5.1. Psychometric Properties of the DASS-21 for Chinese College Freshmen

As for the evaluation of psychometric properties, the reliability of the Chinese version of the DASS-21 was deemed to be acceptable for both samples of college freshmen used. All three scales and the total scores displayed an acceptable degree of internal consistency, which means that the DASS-21 can be reliably used to assess freshmen’s psychological distress. The results obtained in the current study are in agreement with those obtained in previously conducted studies using the Chinese and other versions of the measure for college students [10,11,14]. As for the validity evaluation, the present study adopted a CFA approach to assess factorial validity, and it was demonstrated that the scale fit the one-factor and three-factor structures well in both samples; however, the three-factor structure was a better fit than the one-factor structure. This finding was in line with that of previous studies [10,69]. Moreover, comparing the convergent validity of both samples, the DASS-21 was found to have unsatisfactory discriminant validity. Consequently, we recommend that the DASS-21 be treated as an overall construct of psychological distress, rather than three separate constructs.

### 5.2. Significant and Positive Association between the DASS-21 Results and PSU, PSMU, and PG

Regarding the association between the three PIUs (PSU, PSMU, PG) and freshmen’s psychological distress, the present study adopted SEM analysis. The findings demonstrated that PSU, PSMU, and PG all had a significant and positive influence on freshmen’s psychological distress, which also echoes the results of previous studies [23,29,46,61]. These findings affirm the association in problematic smartphone, social media, or Internet gaming use, in terms of increased likelihoods of suffering from psychological distress. We also found that PSU and PG caused more severe psychological distress for freshmen than PSMU, which is similar to that of prior findings [69]. Despite the fact that no cause-and-effect relationship was established in this study, we speculate that, to some degree, social media platforms can be used by freshmen to maintain communication with family and friends, and also to alleviate loneliness and boredom, both of which can reduce anxiety and long-term distress, and are therefore can be helpful to students, especially when in isolation, as a means of reducing psychological distress [70]. However, we should also note that once an individual has become accustomed to communicating online, rather than face-to-face, the individual may have difficulty communicating effectively offline, which may exacerbate their psychological problems.

### 5.3. Discrepancies between Two Samples Due to the Impact of the Pandemic

Furthermore, it should be pointed out that, based on the fact that the DASS-21 results were found to possess characteristics of measurement invariance across the two samples, *t*-tests could be utilized to evaluate discrepancies between these groups without confounding the interpretation of the items. The present study also found that the state of the mental health of freshman in 2022 was worse that of the 2020 cohort. Especially in terms of depression, freshmen in 2022 were found to be more severely affected. The findings regarding the relatively poorer state of the mental health of the freshmen in 2022 are consistent with that of previous studies conducted on university students [71,72]. Chinese college students were found to experience a higher prevalence of depression (26.0%) during the COVID-19 pandemic, while this prevalence was lower (23.8%) in the pre-COVID-19 era [71]. Peng et al. suggested that college students’ level of anxiety during lockdown conditions increased from June 2020 to June 2021 [72]. In line with these findings, this study revealed that COVID-19 prevention and control policies may serve as potential risk factors for the deterioration of the psychological well-being of college students.

At the end of 2019, the COVID-19 pandemic first broke out in Wuhan, China, but was largely under control by April 2020, across the country [72]. Chinese college students began to return to campus starting in May 2020 [73]. The freshmen in Sample 1 entered college and simultaneously started a new phase of their lives (i.e., in September 2020), after passing their college entrance examination. At that time, the Sample 1 participants did not experience much psychological pressure due to zero (or very few) pandemic cases in their cities. In comparison, Sample 2 participants took part in the current study in the second half of 2022, when the pandemic situation had begun to once again become severe in China. Many colleges and universities quickly launched pandemic prevention and control measures in response to this situation [74], and campuses implemented school closure management and launched fully online teaching (e.g., restricting students from leaving campus, strictly limiting campus activities, and even requiring students to stay in their dorms during emergency situations) [74]. Although these measures effectively protected students from the infection, their harm to the students’ mental health should not be ignored. Indeed, it was more challenging for freshmen, who had just entered university after experiencing an intense high school life and highly competitive college entrance examinations. The situation they faced was not the freedom they expected but, instead, strict lockdown and life in isolation in their dorms. These were huge obstacles for them to adapt to, which could have easily caused them to experience increased psychological distress. Consequently, the mental health statuses of these freshmen, who had recently left their parents, deserves more attention and the monitoring of freshmen students’ psychological distress.

A higher degree of correlation among PSU, PSMU, PG, and psychological distress (i.e., depression, anxiety, and stress) was noted in Sample 2 (October 2022) as compared to Sample 1 (October 2020). This is partly due to the changes in the pandemic situation and in the prevention and control measures. Compared with the second half of 2020, more Chinese colleges and universities had implemented campus closure policies in the second half of 2022 [75]. One prior study demonstrated how people’s daily lives can be adversely affected by prolonged pandemic lockdowns that limit their regular activities, including work and study habits [70]. Students were more susceptible to be afflicted by PIUs when universities were required to cease all campus activities and move all academic activities online as a result of the COVID-19 lockdown measures [76], and the higher severity and prevalence of PIUs, coupled with prolonged lockdowns, has led to poorer mental health.

### 5.4. Summary of Findings and Limitations

To summarize, the results of the present study have two similarities with previous studies. Firstly, as for the DASS-21 factor structure, both one-factor and three-factor structures were found to be acceptable, but the latter was found to be a better fit. This finding echoes that of previous studies [10,68,77]. Secondly, PSU, PSMU and PG were all confirmed to be positively correlated with the depression, anxiety, and stress levels of Chinese college freshmen, which is in line with the findings of previous studies [23,29,46,61,78]. The association was stronger for the data of Sample 2, which shows that college students’ mental health worsened due to the pandemic. This finding is inconsistent with that of a previous study [79]. In addition, the present study also found that the correlation among the three subscales of the DASS-21 was very high, which indicates a low discrimination rate. This finding is different from that of a previous study conducted with Chinese college students, which pointed out that the DASS-21 had good discriminant validity [10].

There are some limitations of this study which must be addressed. First, the method of convenience sampling was adopted in order recruit participants; hence, generalizing the results reported in this study to all Chinese college freshmen would be inappropriate. In addition, the first sample we recruited was smaller than the second. Larger and more representative samples are suggested for future research in order to confirm the preliminary results obtained by the current study. Second, even though we used measurement invariance as the basis for the comparison of the means of the two datasets, we found that there was a significant difference between the two. However, because we did not track the same group of freshmen with a longitudinal design, these conclusions must be taken with caution. More longitudinal studies are necessary in the future.

## 6. Conclusions

According to the findings of the current study, the Chinese DASS-21 is demonstrated to be valid and reliable for assessing the psychological distress of Chinese college freshmen. In addition, the study found that PIUs and psychological distress were significantly correlated, which further confirmed the current concerns centering around the problematic Internet use of Chinese college and university students, especially with regard to PSU and PG.

The findings of this study reiterate the importance of paying close attention to the online behaviors of college freshmen. Moreover, given that freshmen faced many unfamiliar situations while experiencing unexpected lockdowns after they entered their universities, university faculty and healthcare providers should provide proper psychological counseling, and the relevant departments should give more consideration to their mental health when formulating and adjusting pandemic prevention policies.

## Figures and Tables

**Figure 1 healthcare-11-00700-f001:**
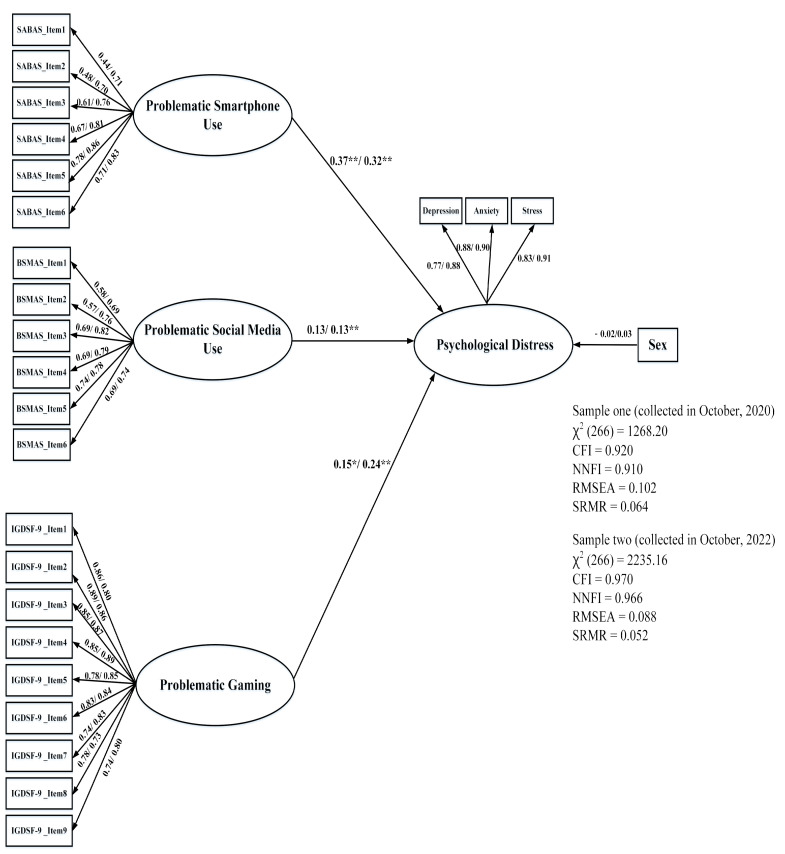
Structural equation modeling of the associations between problematic smartphone use, problematic social media use, and problematic gaming, using the DASS-21. The former and latter values were calculated using the data sets from October 2020, and October 2022. * means *p* < 0.05, ** means *p* < 0.01.

**Table 1 healthcare-11-00700-t001:** Demographic characteristics of study participants.

	Sample 1 (October 2020) *n* = 364	Sample 2 (October 2022) *n* = 956
Age in years; mean (*SD*)	18.17 (0.42)	18.38 (1.11)
Gender; *n* (%)		
Male	116 (31.9)	457 (47.8)
Female	248 (68.1)	499 (52.2)
Department; *n* (%)		
Science	0	151 (15.8)
Literature	120 (33.0)	152 (15.9)
Management	51 (14.0)	156 (16.3)
Engineering	143 (39.3)	274 (28.7)
Others	50 (13.7)	223 (23.3)
Study program level; *n* (%)		
Undergraduate	364 (100)	642 (68.1)
Junior college		301 (31.9)

**Table 2 healthcare-11-00700-t002:** Descriptive and correlation values between the variables.

Variable (Range)	M	*SD*	1	2	3	4	5	6
1. Depression (0–42)	5.33/6.57	5.62/7.38	1					
2. Anxiety (0–42)	7.10/7.06	4.60/6.53	0.62/0.79	1				
3. Stress (0–42)	8.56/8.55	5.71/7.23	0.59/0.80	0.74/0.83	1			
4. PSU (6–36)	17.98/18.92	5.01/6.43	0.37/0.47	0.38/0.46	0.33/0.48	1		
5. PSMU (6–30)	12.85/14.07	3.71/4.46	0.28/0.42	0.30/0.42	0.31/0.44	0.47/0.61	1	
6. PG (9–45)	13.93/17.11	5.46/6.92	0.29/0.48	0.26/0.43	0.22/0.43	0.31/0.50	0.29/0.52	1

Notes: The former and latter values were calculated using data sets from October 2020 and October 2022, respectively. The severity of depression, anxiety, and stress was then assessed using the Depression, Anxiety, and Stress Scale—21-item version where the values were computed by multiplying the score by two; PSU = problematic smartphone use; PSMU = problematic social media use; PG = problematic gaming. All *p* < 0.01.

**Table 3 healthcare-11-00700-t003:** Model fit of the different factor structures.

	χ^2^ (*df*)	CFI	NNFI	RMSEA (90% Confidence Interval)	SRMR	AIC
One-factor model						
Sample 1 (October 2020)	568.67 (189)	0.968	0.965	0.074 (0.067–0.082)	0.091	652.67
Sample 2 (October 2022)	959.83 (189)	0.989	0.988	0.065 (0.061–0.070)	0.051	1043.83
Three-factor model						
Sample 1 (October 2020)	481.79 (186)	0.975	0.972	0.066 (0.059–0.074)	0.086	571.79
Sample 2 (October 2022)	736.94 (186)	0.992	0.991	0.056 (0.052–0.060)	0.046	826.94

Notes: CFI = comparative fit index; NNFI = non-normed fit index; RMSEA = root mean square error of approximation; SRMR = standardized root mean square residual; AIC = Akaike information criterion.

**Table 4 healthcare-11-00700-t004:** Results of convergent and discriminant validity analysis of the DASS-21.

	Depression	Anxiety	Stress
Depression	**0.73/0.84**		
Anxiety	0.92/0.92	**0.58/0.75**	
Stress	0.82/0.93	0.99/0.98	**0.64/0.76**

Notes: Diagonal elements in bold are square root of averaged variance extracted. When these values were higher than the inter-latent factor correlations (off-diagonal elements), discriminant validity was supported for the respective latent variable.

**Table 5 healthcare-11-00700-t005:** Fit indices for measurement invariance across freshmen from two data sets.

	Configural Model	Loadings Constrained as Equal	Loadings and Thresholds Constrained as Equal
One-factor model			
χ^2^(*df*) or Δχ^2^(Δ*df*)	1452.03 (378)	75.36 (20)	518.78 (20)
CFI or ΔCFI	0.987	**−0.001**	**−0.006**
RMSEA or ΔRMSEA	0.066	**0**	**0.011**
SRMR or ΔSRMR	0.091	**0.011**	**0.009**
Three-factor model			
χ^2^(*df*) or Δχ^2^(Δ*df*)	1141.56 (372)	61.86 (18)	187.27 (18)
CFI or ΔCFI	0.989	**0.001**	**−0.002**
RMSEA or ΔRMSEA	0.056	**0**	**0.004**
SRMR or ΔSRMR	0.086	**0.003**	**0.003**

Notes: CFI = comparative fit index; RMSEA = root mean square error of approximation; SRMR = standardized root mean square residual. The **bold values** indicate invariance, i.e., ΔCFI > −0.01; ΔRMSEA < 0.015; ΔSRMR < 0.03 (for factor loading) or <0.01 (for item intercept).

## Data Availability

The data generated for the present study are available from the corresponding authors on reasonable request.

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
