# Peer review of "The Psychometric Properties of the DASS-21 and Its Association with Problematic Internet Use among Chinese College Freshmen"

_healthcare, 2023, doi:10.3390/healthcare11050700_

Round 1
Reviewer 1 Report
The manuscript needs improvement in aspects such as the following:
- Clarify the abstract to simplify the study and the main results.
-In the introduction, make sure the paragraphs are connected. The same ideas are presented repeatedly and there is no consistency.
- Missing information in the process.The procedure for selecting participants (inclusion and exclusion criteria) has not been clarified.
-The sample is so low, justify the reason.
- Please add information on the recruitment of participants and if language was an inclusion / exclusion criterion.
-It is recommended to clarify the discussion section.
-The conclusions section needs to be improved. Clarify ideas presented more consistently.
-It is recommended to use recent citations to support the study
Reviewer 2 Report
I reviewed your study of the reliability and validity of the DASS-21 scale for college freshmen. I found it to be a good job overall. I saw that the analysis results met the appropriate and necessary cut-off values. Congratulations for this work.
I would recommend that you do not just have the limitations section under a separate heading, that you need to provide more resources in the literature, and finally, divide the literature into subheadings according to topics. Thus, it will be easier to read.
-You need to provide more resources in the literature,
- Change the title Research Background with Literature Background.
- Write the literature background and Introduction titles separately.
- Divide the literature into subheadings according to topics. Thus, it will be easier to read.
- Did you conduct exploratory factor analysis (EFA) in the first study?
- Edit the Discussion section more categorically. In other words, make subtitles according to the research questions you asked before and evaluate the results according to the current literature and your own thoughts.
- I think you should take the Limitations under discussion and make a separate topic.
- Explain some implications more understandably in the conclusion part.
Reviewer 3 Report
The current paper describes the psychometric characteristics of the DAS-21 scale in two samples of college freshman from China, and attempts to verify the presence of PIU and the relationship of it with other measures of psychological suffering, such as depression. The methods are clear and adequate, and results are adequately drawn from it.
The major limitation of this study is to be based on a convenience sample. This most probably won't affect the psychometric characteristics of the scale, tough. And this is acknowledged as a limitation.
I suggest an extensive review of the English in the text.
